# Comprehensive Evaluation of Quality Characteristics of Four Oil-Tea Camellia Species with Red Flowers and Large Fruit

**DOI:** 10.3390/foods12020374

**Published:** 2023-01-13

**Authors:** Shengyue Zhong, Bin Huang, Teng Wei, Zeyuan Deng, Jing Li, Qiang Wen

**Affiliations:** 1State Key Laboratory of Food Science and Technology, Nanchang University, Nanchang 330047, China; 2Jiangxi Provincial Key Laboratory of Camellia Germplasm Conservation and Utilization, Jiangxi Academy of Forestry, Nanchang 330047, China

**Keywords:** red-flowered oil-tea camellia, *Camellia chekiangoleosa* oil, correlation analysis, gray correlation coefficient, triglycerides

## Abstract

Red-flowered oil-tea camellia (*ROC*) is an important woody oil species growing in the south, and its oil has high nutritional value. There are four main species of *ROC* in China, namely, *Camellia chekiangoleosa* (*CCH*), *Camellia polyodonta* (*CPO*), *Camellia semiserrata* (*CSE*) and *Camellia reticulata* (*CRE*). Reports on the comprehensive comparative analysis of *ROC* are limited. This study investigated the fruit characteristics and nutritional components of four *ROC* fruits, and the results showed that *ROC* had high oil content with levels of 39.13%–58.84%, especially the *CCH* fruit, which reached 53.6–58.84%. The contents of lipid concomitants of *ROC* oil were also substantial, including β-amyrin (0.87 mg/g–1.41 mg/g), squalene (0.43 mg/g–0.69 mg/g), β-sitosterin (0.47 mg/g–0.63 mg/g) and α-tocopherol (177.52 μg/g–352.27 μg/g). Moreover, the transverse diameter(TD)/longitudinal diameter (LD) of fruits showed a significant positive correlation with the oil content, and *ROC* fruits with thinner peels seemed to have better oil quality, which is similar to the result of the oil quality evaluation obtained by the gray correlation coefficient evaluation method. Four *ROC* oils were evaluated using the gray correlation coefficient method based on 11 indicators related to the nutritional value of *ROC*. *CCH* oil had the highest score of 0.8365, and YS-2 (a clone of *CCH*) was further evaluated as the best *CCH* oil. Finally, the results of heatmap analysis showed that triglycerides could be used as a characteristic substance to distinguish *CCH* oil from the other three *ROC* oils. The PLSDA (Partial least squares regression analysis) model and VIP (Variable important in projection) values further showed that P/S/O, P/O/O, P/L/L, P/L/Ln, S/S/O, S/O/O and P/S/S (these all represent abbreviations for fatty acids) could be used as characteristic differential triglycerides among the four *ROC* oils. This study provides a convenient way for planters to assess the nutritional quality of seed oil depending on fruit morphology and a potential way to distinguish between various *ROC* oils.

## 1. Introduction

The oil of *Camellia* spp. has become a well-known high-quality edible oil because of its rich nutrition [1]. For example, *Camellia oleifera Abel.*, one of the most important ligneous edible trees, originated in China. Red-flowered oil-tea camellia (*ROC*) usually refers to oil-tea camellia with red flowers and large fruit in *Sect. Camellia* plants of the genus *Camellia*. Compared with *C. oleifera*, *ROC* has higher economic value and development prospects because of its unique oil quality and bright colors [2]. Although the planting area of *ROC* in China is less than that of *C. oleifra*, *ROC* has higher oil content, in general. Nowadays, *ROC* has become an important germplasm resource for oil-tea *camellia* cultivar breeding. Due to their high oil value, there are many common species of *ROC*, including *Camellia chekiangoleosa Hu. (CCH)* (distributed in the mountains of Zhejiang, Jiangxi, Hunan, Hubei, southern Anhui and northern Fujian at altitudes of 600–1400 [3]), *Camellia semiserrata Chi. (CSE)* (mainly distributed in Guangdong and parts of Guangxi), *Camellia polyodonta How. ex. Hu. (CPO)* (specific to Guangxi species) and *Camellia reticulata Lindl. (CRE)* (with the largest distribution in the range of 1700–2100 m above sea level in Yunnan), which have a certain cultivation area in China. *CCH* is the major species of *ROC* because it has a short reproductive cycle (it only takes six months from flowering to ripening), a high oil yield, stable biological heritability and the most extensive planting area among the four *ROC* species, and it has become a more valuable research object and ideal experimental raw material in recent years [4].

At present, the research on *ROC* at home and abroad is mainly associated with the quality of *ROC* oil, focusing on its antioxidant properties, aroma components and the influence of processing methods on oil quality. The oil of *CCH*, containing about 90% unsaturated fatty acids [5], which mainly are oleic acid [6], is beneficial to human health and can be used as advanced natural skin care cosmetics due to its high oxidation stability [7]. High levels of unsaponifiable substances, such as squalene, phytosterols, phenolic compounds, α-tocopherol and volatile compounds, in addition to a high content of oleic acid, are characteristic of *CCH* oil, which is helpful in lowering blood cholesterol and triglycerides, thereby preventing high blood pressure [8], heart disease, arteriosclerosis and other diseases [9]. The unsaturated fatty acids in *CSE* oil account for 85.312%, the iodine value is 84 g I_2_/100 g oil, and the storage period can reach 340 days [10]; its unsaturated fatty acid content is higher than that of olive oil, and its various oil indicators are first-class and more resistant to storage, so it has good market development prospects. The average content of linoleic acid in *CPO* is 8.73%, which is in line with the standard of healthy, nutritional oils and fats stipulated by the World Health Organization (WHO) [6]. *CRE* oil is a kind of high-quality and storage-resistant edible oil; its color is clear, and it is fragrant and rich in nutrients. The content of unsaturated fatty acids is about 90% and includes ample vitamin E and various mineral elements. It is worth noting that the polyphenols in *CRE* oil have a strong antioxidant capacity [11]. However, these studies on *ROC* oils were isolated and disorganized and did not provide a comprehensive comparative analysis of the four main *ROC* oils.

Furthermore, the existing literature shows that almost all studies on *ROC* fruits focus on oil quality. However, evaluation methods based on the final oil quality are not applicable for camellia tree planters. Therefore, the present study tried to explore the relationship between fruit characteristics (such as fruit morphology and peel chemical composition) and the final oil quality, aiming to provide a convenient way for planters to assess the nutritional quality of seed oil depending on fruit morphology.

For this purpose, this study investigated the fruit characteristics and the contents of macronutrients and antioxidant substances in the peels and kernels of four *ROC* fruits in order to explore the correlation between different parts of the fruit to provide a theoretical basis for comprehensively understanding the nutritional quality of camellia seed oil. The fatty acid composition and levels of lipid concomitants of four *ROC* oils were also measured, aiming to explore the nutritional value of various *ROC* oils. The gray correlation coefficient method was employed in an accurate and comprehensive evaluation of oil quality, with the aim of picking out the best-quality oil. The triglyceride composition is also an important characteristic of the oil, and its determination will reveal the essential structural properties of *ROC* oils, which provides a potential way to distinguish between various *ROC* oils.

## 2. Materials and Methods

### 2.1. Materials

*ROC* test materials were all provided by the Jiangxi Academy of Forestry Science and were collected from the same forest area in Leping, Jiangxi (28.58° N, 117.08° E). The test *CCH* fruit materials were selected from 5 clones of different genotypes (YS-1, YS-2, YS-4, LK005 and LP22, respectively), and *CPO*, *CSE* and *CRE* fruit materials were mixed samples of 5 clones. The fruits were all hand-harvested in the fruit maturity period. The fresh fruits were sealed in a ziplock bag and brought back to the laboratory. After being dried to constant weight at 50 degrees and unshelled, the seeds were dehusked by hand or a hammer, and the sample was crushed using a grinder (Hong Jing Tian, DE-300 g, Jinhua, China). The seed and peel powders were stored at 4 °C until the analysis process.

### 2.2. Chemicals and Standards

Glucose, squalene, α-tocopherol, gallic acid, catechins, β-amyrin and β-sitosterol standards were obtained from Yuanye Bio-Technology Co., Ltd. (Shanghai, China). Methyl acetate and oxalic acid were obtained from Aladdin Industrial Corporation (Shanghai, China). n-Hexane, acetonitrile, ammonium formate, formic acid and methanol (purchased from Honeywell International (Shanghai, China)) were of chromatographic reagent grade, and other solvents were of analytical reagent grade (Shanghai, China).

### 2.3. Determination of Morphological Index of ROC

According to Lu et al. [12], the longitudinal diameter, transverse diameter and peel thickness of *ROC* fruits in each variety were randomly measured using vernier calipers, while the total masses of fresh fruits, fresh seeds, dried seeds and dried kernels were weighed using a balance to obtain the fresh seed rate and kernel rate, and the quantity of all varieties measured was greater than or equal to 20 kg. The fresh seed rate is expressed as the ratio of the total mass of fresh seeds to the total mass of fresh fruit (%). Similarly, kernel yield is expressed as the ratio of dry kernel weight to dry seed weight (%).

### 2.4. Analysis of Macronutrients in ROC Fruit

#### 2.4.1. Moisture Content Analysis

According to the methods of Wei et al. [13], moisture content was determined from the change in weight before and after drying fresh peels and camellia seed kernels to constant weight. Fresh samples were dried in an oven at 105 °C until constant weight.

#### 2.4.2. Oil Content Analysis

The determination of oil content was carried out using the method of Bouali [14] (with slight modifications). One gram of dried sample powder was put in a Soxhlet extractor, and then petroleum ether (30–60 °C) was added to it to extract for 8 h at 50 °C. Then, the extraction bottle was dried at 105 °C to constant weight. The oil content (%) was the ratio of the mass of the pre- and post-extraction bottles to the mass of the extracted substances.

#### 2.4.3. Soluble-Sugar and Starch Content Analysis

Soluble sugars were extracted from 0.1 g of de-oiled residue according to the method of Luo [15], and then the remaining residue was added to hydrochloric acid according to GB 5009.9-2016 to acidify the starch into soluble sugars. The soluble-sugar and starch contents were determined using the anthrone colorimetric method. Results are expressed as a percentage of dry weight (% dry weight).

#### 2.4.4. Protein Content Analysis

First, 0.5 g of dried sample powder, 0.2 g of potassium sulfate, 3 g of copper sulfate and 10 mL of sulfuric acid (98%) were added to the digestion tube and digested in a digester (K9860, Haineng Future Technology Group Co., Ltd., Jinan, China) for 5 h. After the digestion, the digestion solution was distilled into ammonia gas with sodium hydroxide solution, and the ammonia gas was absorbed by 2% boric acid. After blank correction [16], it was titrated with 0.1 mol/L hydrochloric acid to determine the nitrogen content. Results are expressed as a percentage of dry weight (% dry weight).

#### 2.4.5. FCR-Reducing Capacity and Total Flavonoid Content Analysis

The dried sample was weighed (0.1 g) and then dissolved in 3 mL of 70% methanol aqueous solution as the extraction solution. According to the study by Pawels [17] and Huang [18], FCR measures the reducing capacity of samples and does not reflect the total phenolic content, so the FCR-reducing capacity was determined using the Folin–Ciocalteu method [19] (slightly modified), with gallic acid as the standard. For this test, 25 μL of the standard and sample solutions and 125 μL of Folin phenol reagent were added to a 96-well plate and reacted for 10 min, and then 100 μL of 7.5% sodium carbonate solution was added. The sodium carbonate solution was shaken on a shaker (SCILOGEX, SLK-O3000-S, Shanghai, China) for 30 min and then measured at 765 nm. Total flavonoids were determined by the sodium nitrite–aluminum trichloride method. First, 110 μL of sodium nitrite (0.066 M) was mixed with 25 μL of the standard catechin or sample solution for 5 min, and then it was reacted with 15 μL of aluminum chloride (0.75 M) solution and 100 μL of sodium hydroxide (0.5 M) solution and finally measured at 510 nm. Results are shown as percent by weight of dry sample (μg/g).

### 2.5. Analysis of Lipid Concomitant Content of ROC Oil

#### 2.5.1. Oil Extraction

The method for extracting oil from the seed kernel was consistent with the method for determining oil content. After the Soxhlet extraction was completed, the petroleum ether was removed via vacuum spin evaporation and nitrogen, and the obtained oil samples were stored at −20 °C for the determination of inhibitory concomitants, fatty acid composition and triglycerides.

#### 2.5.2. Determination of α-Tocopherol

About 0.1 g of oil was dissolved in n-hexane and filtered through a 0.2 μm membrane. Then, 10 μL of the sample was injected into an Agilent 1100 series HPLC apparatus equipped with a diode array detector (DAD) and a Hypersil ODS2 column (4.6 mm × 150 mm, 5 µm, Elite, Dalian, China). The mobile phase was 96% methanol and 4% water, the flow rate was 0.8 mL/min, the column temperature was 25 °C, and the detection was performed at 292 nm [20]. Quantitative analysis of α-tocopherol was carried out using the external standard method.

#### 2.5.3. Determination of Sterols and Squalene

Approximately 0.1 g of oil sample and 100 μL of 1 mg/mL 5α-cholestanol (internal standard) were saponified with 5 mL of 2 moL/L potassium hydroxide–ethanol solution and vortexed for 60 s in a 75 °C water bath for 30 min. Then, 5 mL of n-hexane and 3 mL of distilled water were added to the saponified sample and then centrifuged at 4500 r/min for 5 min, and the supernatant was extracted with n-hexane and derivatives at 65 °C for 1 h. Next, 1 μL of the sample was injected into the GC-MS (Gas chromatographic mass spectrometry)instrument, and the temperature program was 200 °C for 0.5 min, rising to 300 °C at 10 °C/min and held for 20 min. The inlet temperature was 280 °C, the oven temperature was 200 °C, the carrier gas was hydrogen, the carrier gas flow rate was 1.2 mL/min, and the split ratio was 50:1. Quantitative analysis of sterols [21] and squalene [22] was carried out using internal and external standard methods, respectively.

### 2.6. Determination of Fatty Acid Composition

An oil sample of 2–3 mg was dissolved in 1.5 mL of n-hexane, methylated with 40 μL of methyl acetate and 100 μL of sodium methoxide–methanol solution for 30 min [23], and then analyzed using gas chromatography (6890 N, Agilent Technologies, Santa Clara, CA, USA).

Nitrogen was used as the carrier gas; hydrogen and air were used as combustion gases at a flow rate of 1.8 mL/min. Gas chromatography was performed using a flame ionization detector and a CP-Sil88 column (CP7489, 100 m × 0.25 mm × 0.2 μm, Chrompack; Agilent, Santa Clara, CA, USA) at an inlet temperature of 250 °C and operated in splitless mode. The flame ionization detector’s (FID’s) temperature was 250 °C; the column pressure was 24.52 psi. The initial column temperature was 45 °C, which was maintained for 5 min, then ramped to 175 °C at 13 °C/minute and held for 27 min, and finally programmed to 215 °C at 4 °C/minute and held for 35 min. Fatty acids were identified by comparing their retention times to that of a standard FAME mix (#463) (GLC463, NuChek. Prep, Minneapolis, MN, USA).

### 2.7. Triglyceride Analysis

About 3 mg of oil was dissolved in 1 mL of n-hexane and passed through a 0.22 μm organic filter membrane for Q-TOF [24] analysis (high-performance benchtop quadrupole-orbitrap LC-MS/MS system). Liquid chromatography conditions were as follows: Zorbax Eclipse plus C18 column (2.1 mm × 50 mm × 1.8 μm) was used. Mobile phase A was an acetonitrile (60%) and water (40%) solution, mobile phase B was an acetonitrile (10%) and isopropanol (90%) solution, and mobile phases A and B were both added with 0.1% formic acid and 10 mmol/L ammonium formate. The elution conditions were 0–1 min 70% A, 1–31 min 87% A, 31–32 min 87% A, 32–33 min 70% A and 33–37 min 70% A. The flow rate was 0.2 mL/min, the injection volume was 2 μL, and the column temperature was 45 °C. Mass spectrometry conditions were an ESI ionization source, positive ion mode, collision energy of 20 v and fragmentation energy of 150 v.

### 2.8. Gray Correlation Coefficient Method

The gray correlation coefficient method was performed in accordance with Deng. It is an evaluation method that measures the degree of association between data by studying the size of the data association. According to the quality meaning of the selected index, the optimal value of each index was selected as the ideal data series $x_{0}\left(k\right)\left(k=1,2,3\cdots ,n,\;representing\;n\;quality\;indicators\right)$ and initialized with the formula $x_{i}^{’}\left(k\right)=\frac{x_{i}\left(k\right)}{x_{0}\left(k\right)}\left(x_{0}\left(k\right)\;bigger\;is\;better\right)$.

The formula $\Delta_{i}\left(k\right)=\left|x_{0}^{’}\left(k\right)-x_{i}^{’}\left(k\right)\right|$ was used to find the absolute difference, and the correlation coefficient can be obtained by the formula $\delta_{i}\left(k\right)=\frac{min\Delta_{i}\left(k\right)+\rho max\Delta_{i}\left(k\right)}{\Delta_{i}\left(k\right)+\rho max\Delta_{i}\left(k\right)}$, where *ρ* is the resolution coefficient and is generally taken as 0.5. Each correlation coefficient was substituted into the formula $r_{i}=\frac{1}{n}\sum_{i=1}^{n}\delta_{i}\left(k\right)$ to calculate the correlation degree of each sample. Then, the weight of each indicator was calculated according to the formula $W_{k}=\frac{r_{i}}{\sum r_{i}}$; finally, the weight was substituted into the comprehensive evaluation formula $r_{i}^{’}=\sum_{i=1}^{n}W_{k}\delta_{i}\left(k\right)$, and the comprehensive score of the sample was calculated.

### 2.9. Statistical Analysis

All experiments were performed in parallel in three groups, and the experimental data are expressed as mean ± standard deviation. All statistical analyses were performed using Statistical Product and Service Solutions (SPSS) 23.0 (IBM, Armonk, NY, USA). Picture drawing and Pearson correlation analysis were performed using GraphPad Prism 9.0 (GraphPad Software Inc., San Diego, CA, USA). Heatmaps and PLSDA models were analyzed through the metaboanalyst website (https://www.metaboanalyst.ca/, accessed on 16 September 2022). The standard for screening differential glycerides was a *p* value less than 0.05 and a variable importance in projection (VIP) value > 1.

## 3. Results and Discussion

### 3.1. Phenotype, Fresh Seed Rate and Kernel Rate of Four ROC Fruits

The phenotype of *ROC* included the longitudinal diameter (LD), transverse diameter (TD) and peel thickness (PT). As shown in Table 1, the successive order of average fruit size is *CSE > CPO > CRE > CCH.* The apparent shape of *CSE* was the largest with the maximum LD (117 ± 0.40 mm), TD (108 ± 0.42 mm) and PT (28 ± 0.14 mm), followed by *CPO* with an LD of 107 ± 0.82 mm, a TD of 79 ± 0.52 mm and a PT of 21 ± 0.98 mm. However, the apparent shapes of *CCH* and *CRE* were significantly smaller than those of *CSE* and *CPO* (*p* < 0.05), which is consistent with the results of Shen [25].

The seed refers to the hard-shelled fruit of the *ROC* fruit after being peeled, and the kernel refers to the seed after being shelled. *CCH* showed the highest fresh seed rate of 20.75% (LK005)-34.40% (YS-2), followed by *CRE* (22.5%), *CPO* (14.12%) and *CSE* (9.32%). Likewise, the seed kernel rate of *CCH* (60.31% (YS-4)-65.62% (LK005)) ranked first among the four *ROC* oils, followed by *CRE* (58.50%), *CPO* (50.41%) and *CSE* (49.13%). Evidently, the fresh seed rate and seed kernel rate seemed to be positively affected by the TD of fruits, while they decreased with the increase in the fruit’s LD (Figure 1). So, the ratio of TD to LD positively influenced the fresh seed rate and seed kernel rate (Figure 1). Furthermore, the PT of fruit possibly restricted the development of seeds too, which was demonstrated by a strong negative relationship between the PT and the fresh seed rate and seed kernel rate. All in all, fatty seeds (high ratio of TD to LD) with thin peels generally mean a high fresh seed rate and seed kernel rate, which might affect the final oil quality.

### 3.2. Chemical Compositions of ROC Kernel/Peel Depending on Different Species

The chemical compositions of *ROC* seeds/peels are shown in Table 2. The moisture of fresh peel ranged from 75.72% (*CCH*) to 82.10% (*CPO*). Starch (118.61 mg/g (CRE)–127.59 mg/g (*CPO*)) was the most abundant substance in dried *ROC* peel, followed by soluble sugars (48.20 mg/g (*CPO*)–128.08 mg/g (*CRE*)), protein (22.6 mg/g (*CPO*)–34.3 mg/g (*CRE*)), oil (26.77 mg/g (*CCH*)–40.33 mg/g (*CRE*)), FCR-reducing capacity (4.46 mg/g (*CRE*)–10.31 mg/g (*CSE*)) and total flavonoids (1.94 mg/g (*CRE*)–5.32 mg/g (*CSE*)). Among the five clones of *CCH*, the peel of YS-1 was rich in protein (47.24 mg/g), starch (125.71 mg/g) and flavonoids (4.32 mg/g), while the YS-4 peel had abundant soluble sugars (10.31 mg/g) and starch (5.32 mg/g), and then LP22 showed a higher content of starch (132.41 mg/g), oil (28.9 mg/g), FCR-reducing capacity (10.36 mg/g) and flavonoids (5.47 mg/g). It is worth mentioning that such oil and protein have been seldom reported in the peel; although their contents were relatively low, their value to the human body is self-evident, and the development and utilization of the protein and oil in the *ROC* peel may provide us with a new research direction and hotspot.

On the contrary, the contents of substances in the kernel were quite different from those in the peel. The moisture in the kernel was much less than that in the peel and ranged from 21.87% (*CCH*) to 45.33% (*CPO*); however, the protein, soluble sugars, oil and flavonoids contained in *ROC* kernels were 2–3 times, 1.5–3 times, 10–20 times and 2–6 times those in the peel, respectively. Among the four varieties of *ROC*, the kernel of *CPO* was rich in soluble sugars (178.84 mg/g), starch (84.37 mg/g) and flavonoids (12.58 mg/g), while the kernel of *CCH* contained more protein (91.87 mg/g) and oil (564.41 mg/g) and had a higher FCR-reducing capacity (9.02 mg/g). In the dried *CCH* kernel, LP22 showed a higher protein content (105.86 mg/g), oil content (588.36 mg/g) and FCR-reducing capacity (9.35 mg/g) than other clones. Oil was the most essential factor that directly determined the edible benefits of kernels, and the oil content of *ROC* (39.13–58.84%) was higher than that of camellia sinensis oil seeds (26.33–31.81%) [7] and the dominant species of *camellia oleifera* (about 40%). Furthermore, the oil content of *CCH* was particularly dazzling, ranging from 53.6% to 58.84%, which indicates that *ROC*, especially *CCH*, has higher nutritional value as a woody oil crop than *camellia oleifera* and *camellia sinensis* oils. The seed kernels after oil extraction also had high nutritional value; for example, the polysaccharides contained in *camellia oleifera* seeds have a hypoglycemic effect, and the yield can reach 5.93%, while flavonoids and polyphenols have antioxidant effects.

Although the protein, total soluble sugars, total flavonoids and starch in the peel were weakly positively correlated with the oil content of the seed kernel, the oil content in the peel and the oil content of the seed kernel showed an opposite trend, which is shown in Figure 1. There was no doubt that nutritional substances in the peel were transformed in the kernel to provide material to synthesize seed oil, which inhibited the accumulation of peel oil. Moreover, the oil content in the kernel was positively correlated with the seed kernel yield of *ROC*. The positive correlation between the kernel yield and oil content is well understood: because the process of kernel growth involves the transformation of a gel state to a solid state, and this process is accompanied by a decrease in moisture content, the decline in the water phase in *C. chekiangoleosa* seeds creates the conditions for oil accumulation [13]. The relationship between the kernel yield and oil quality is partly consistent with the correlation between fresh seed yield and oil. Both the LD and PT of fruits were negatively associated with the seed oil content. However, opposite results to the LD and PT were observed for the TD of fruits. Moreover, it is worth noting that the ratio of TD/LD showed a strong positive correlation with the oil content of kernels. Both the peel and the kernel are important components of the *ROC* fruit, and the raw material of the peel is easily available in the *ROC* fruit. By studying the relationship between peel nutrients and kernel nutrients, it is possible to monitor the oil content.

### 3.3. Lipid Concomitants of the Oil of Four ROC Oil

The contents of lipid concomitants of the four *ROC* oils are shown in Table 3. β-Amyrin (0.87 mg/g (*CPO*)–1.41 mg/g (*CCH*)) was the most abundant lipid concomitant in *ROC* oil, followed by squalene (0.43 mg/g (*CSE*)–0.69 mg/g (*CCH*)), β-sitosterin (0.47 mg/g (*CSE*)–0.63 mg/g (*CCH*)), α-tocopherol (177.52 μg/g (*CSE*)–352.27μg/g (*CRE*)), *FCR*-reducing capacity (130.72 μg/g (*CSE*)–164.82μg/g (*CPO*)) and total flavonoids (353.49 μg/g (*CSE*)–438.18μg/g (*CCH*)), which agrees with the study by Shi [26]. Compared with other vegetable oils, *ROC* oil contained more β-amyrin (0.61–0.91 mg/g) than *Camellia sinensis* oil (0.36–0.54 mg/g) and more α-tocopherol than sesame oil [27]. Moreover, *ROC* oils had higher contents of squalane (0.43 mg/g–0.72 mg/g), β-sitosterin (0.47 mg/g–0.74 mg/g) and β-amyrin (0.87 mg/g–1.91 mg/g) than *Camellia oleifera* oil (0.12 mg/g–0.25 mg/g, 0.11 mg/g–0.24 mg/g and 0.61 mg/g–0.91 mg/g, respectively) [7]. Obviously, *CCH* oil contained more nutrients than the other camellia oils, especially squalene (0.69 mg/g), β-sitosterin (0.63 mg/g) and β-amyrin (1.41 mg/g). Among the five clones of *CCH*, LP22 oil showed the highest levels of α-tocopherol (243.80 μg/g), FCR-reducing capacity (171.65 μg/g) and total flavonoids (449.27 μg/g).

Lipid concomitants have been reported to be free radical scavengers, which could improve the oxidative stability of oil and delay the oxidation of vegetable oil. Moreover, fat-soluble concomitants of vegetable oil are also associated with health benefits. For example, α-tocopherol is an essential vitamin for the human body, which functions to reduce fat, prevent cardiovascular and cerebrovascular diseases and improve human immunity [28]. Phytosterols are active components widely existing in plants and plant seeds, which are dominated by β-sitosterin [29]. Phytosterols could improve cardiovascular and cerebrovascular diseases, such as atherosclerosis, caused by high low-density lipoprotein cholesterol [30]. Phytosterols are also beneficial to the human body, providing functions such as antioxidation, blood lipid reduction, anticancer and anti-inflammatory activity and immune regulation [8]. Furthermore, the nutritional effects of squalene on the human body have been demonstrated to include a strong oxygen-carrying capacity, which could revitalize the body, promote metabolism and improve the immune function of the body. Moreover, squalene also has antiaging and antitumor effects [31]. Therefore, *ROC* oil contained higher levels of lipid concomitants than others, which might exhibit high oxidation stability and high nutritional value and could be used as a potential raw material in functional foods and medicine. As shown in Figure 1, the characteristics of *ROC* significantly affect the contents of lipid concomitants of the oil. Both the LD and PT of fruits were negatively associated with lipid concomitants, especially squalane. In contrast, like TD, the ratio of TD/LD is positively correlated with the contents of lipid concomitants. The contents of lipid concomitants were also positively correlated with the fresh seed rate and seed kernel rate. This is probably because the oil content of the seed kernel can directly affect the contents of lipid concomitants, and these characteristics of *ROC* affect the contents of lipid concomitants by influencing the oil content of the seed kernel. The health benefits of lipid concomitants are well known, and therefore, this result provides a reference for choosing camellia fruits with higher oil quality according to fruit morphology.

### 3.4. Fatty Acid Composition

As shown in Table 4, oleic acid was the most important fatty acid in *ROC* oil, which accounted for 75.65% (*CRE*)–80.58% (*CSE*) of total fatty acids. High oleic acid in edible oil generally functions to lower cholesterol, delay atherosclerosis, regulate blood lipid levels and improve immune function and antioxidation [32], which implies the nutritional value and broad applicability of camellia oil. *ROC* oil contained low levels of saturated fatty acids, mainly palmitic acid and stearic acid, with contents of 7.51% (*CSE*)–10.88% (*CRE*) and 2.47% (*CPO*)–3.82% (CCH), respectively. Similarly, the contents of linoleic acid and linolenic acid in camellia oil were not high, with levels of 5.77% (CCH)–8.86% (*CPO*) and 0.58% (*CCH*)–1.04% (*CPO*). Obviously, the fatty acid composition of *ROC* oils was very similar; their oleic acid content was higher than that of *camellia sinensis* oil (56.79%) [33] but close to that of *camellia oleifera* oil (78.24%) [34]. However, it is worth noting that the content of linoleic acid in *CCH* oil was relatively low, and conversely, an increase in the proportion of linoleic acid in polyunsaturated fatty acids reduces the stability of camellia oil [35], so compared to the other three *ROC* oils and *camellia oleifera* oil, *CCH* oil was more resistant to preservation. In the present study, the ratio of n-6/n-3 ranged from 7 to 12.98, which complies with the FAO-recommended ratio (5–10:1) [36]. According to Simopoulos A.P.’s [37] study, it is necessary to increase the intake of n-3 PUFAs in the diet to maintain a reasonable dietary ratio of n-6 and n-3 PUFAs, and a lower proportion of n-6/n-3 PUFAs was more effective in the prevention and treatment of chronic diseases. Interestingly, any of several antioxidant substances that react with FCR in the peel seemed to prevent the oxidation of the oil in the fruit, as shown in Figure 1, where the FCR-reducing capacity was positively correlated with the oleic acid content of the oil, but negatively correlated with the C18:2 and C18:3 content. As previously reported, compounds that can react with FCR in plants have extremely strong antioxidant properties [38,39]. Oil in the peel had a significant negative correlation with the C18:1 content by influencing the oil content in the kernel.

### 3.5. Evaluation of the Quality of Four ROC Oils by Gray Correlation Coefficient Method

Gray system theory is widely used in data generation, system control, correlation analysis and other issues. In recent years, many studies have applied the gray system method to quality evaluations. Shen B. [40] compared the quality of three camellia oils through the gray correlation coefficient method and came to the conclusion that the quality of *CCH* oil was the best. Moreover, Ou Mingyi [41] analyzed the main intrinsic qualities of the tobacco leaf formula module using the gray correlation degree.

The gray correlation coefficient method was employed to evaluate the nutritional quality of the four oils in the present study. The indicators selected in the evaluation of *ROC* oil could directly or indirectly reflect the edible characteristics of the fruit. As shown in Figure 2A, β-sitosterol had the highest weight ratio of 16.72%, followed by linolenic acid (11.23%), the seed kernel rate (10.50%), β-amyrin (9.95%), squalene (9.47%), the fresh seed rate (7.96%), oil content (7.94%), a-tocopherol (7.20%), peel thickness (7.10%), oleic acid (5.99%) and linoleic acid (5.94%). The final correlation degree of *ROC* oil is exhibited in Figure 2B. *CCH* oil (the mean values of the five clones were calculated and compared) had the highest score among *ROC* oils (0.8365), followed by *CRE* (0.5879) and *CPO* oils (0.5326), while the lowest score was obtained for *CSE* (0.4213). Interestingly, both *CCH* and *CRE* indeed had a higher TD/LD and a lower PT than *CPO* and *CSE*, which also demonstrated the possibility of using fruit morphology to predict the camellia oil quality. Furthermore, the five clones of *CCH* oil were also similarly evaluated by the gray correlation coefficient, as shown in Figure 2C,D. Linolenic acid had the highest weight value, and the clones of YS-2 and LK005 were evaluated as having the best *CCH* score. Notably, YS-2 and LK005 exhibited a higher TD and a thinner peel. Obviously, the score of *CCH* oil was much higher than those of the other three main *ROC* oils, reflecting the edible function superiority of *CCH*. The scores of YS-2 and LK005 were higher than those of other clones, so YS-2 LK005 deserved more focus for its consumption.

### 3.6. Triglyceride Composition Could Be Used to Identify Four ROC Oils

In addition to having the highest nutritional value among all *ROC* oils, *CHH* oil contained a specific triglyceride composition, which is helpful in identifying high-quality *CCH* oil from other *ROC* oils, even *C. oleifera*. This is beneficial to manufacturers, consumers and researchers when testing for the adulteration of camellia oil. As shown in Appendix A, 132 kinds of triglycerides were identified in Q-TOF-MS/MS (high-performance benchtop quadrupole-orbitrap LC-MS/MS system). Obviously, as a result of the high level of oleic acid, O/O/O was the most abundant triglyceride in camellia oil, which ranged from 36.23% to 43.11%, followed by P/O/O (5.43–8.23%), P/S/O (3.62–5.29%), S/O/O (3.49–5.83%), P/L/L (2.87–4.12%), S/S/O (2.63–5.17%), P/P/O (2.22%-3.83%), L/L/L (2.19–6.08%), O/L/L (1.41–4.03%), O/O/L (1.28–3.31%) and P/O/L (0.54–2.37%), which is similar to Wei’s [42] study.

Triglycerides could be characteristic substances of the four *ROC* oils. As shown in the heatmap (Figure 3A), the five clones of *CCH* oil clustered together and were significantly separated from the other three *ROC* oils, which indicates that triglycerides could be used as markers to distinguish between the four *ROC* oils. As shown in Figure 3B, a PLSDA model was used to screen out the characteristic triglycerides among the four *ROC* oils. In the PLSDA model diagram, the proportions of principal components **1** and **2** were 26.1% and 14.4%, respectively. The four kinds of *ROC* oils were separated from each other, and the five clones of *CCH* oil were grouped together. Seven characteristic triglycerides (P/S/O, P/O/O, P/L/L, P/L/Ln, S/S/O, S/O/O and P/S/S) were screened out by their VIP values (Appendix A) greater than 1 in the PLSDA model and were close to 1%. Based on the seven characteristic triglycerides, the *CCH* oils were separated from each other, and the five clones of *CCH* oil were grouped together, as shown in Figure 3C. *CCH* oil had high contents of P/S/O (4.67%), S/S/O (5.17%), P/O/O (8.23%) and P/L/L (4.12%). As shown in Figure 3D, all characteristic triglycerides were normalized for hierarchical cluster analysis to build a dendrogram, and the results showed that the five clones of *CCH* oil had a clear clustering trend. When 20 distance thresholds were set, the tree-like structure of the cluster analysis results was divided into two main parts, from which *CCH* oil and the other three varieties of camellia oil were clearly classified. Therefore, the seven triglycerides could be potentially used as characteristic lipids to distinguish the four varieties of *ROC* oils, which is important for identifying *CCH* oil with high quality.

## 4. Conclusions

In this study, the nutrient compositions and contents of four *ROC* fruits at maturity were comprehensively determined. The relationship between fruit characteristics and oil quality was predicted from experimental results. Moreover, the results showed that the oil content of the seed kernel in *ROC* was higher than that of *Camellia oleifera*, and the oil content of the *CCH* seed kernel was the highest in *ROC*. Although the fatty acid composition of *ROC* oil and *Camellia oleifera* oil is similar, the contents of lipid concomitants of *ROC* oil are significantly higher than those of *Camellia oleifera* oil; similarly, *CCH* is the most prominent in *ROC* oil. Therefore, the gray correlation coefficient method was used to evaluate the four *ROC* oils. The research shows that *CCH* has the best quality, and the YS-2 clone had the highest score among the clones. Finally, triglycerides can be used as characteristic substances to distinguish *CCH* from the other three *ROC* oils, which provides a theoretical basis for the adulteration analysis of *CCH*.

## Figures and Tables

**Figure 1 foods-12-00374-f001:**
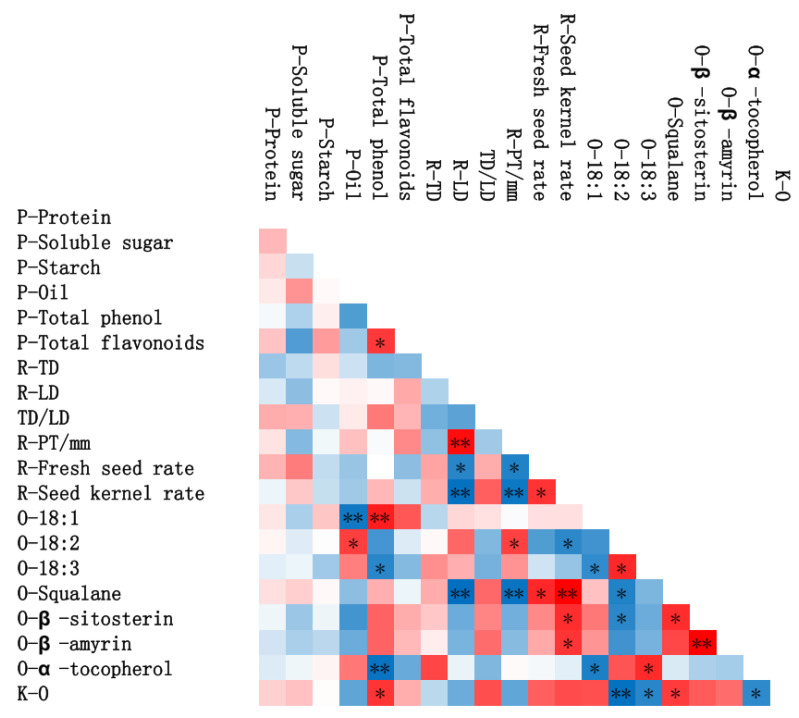
Red represents positive correlation, blue represents negative correlation, and the darker the color, the greater the correlation. * Significant at the 0.05 level (bilateral); ** significant at the 0.01 level (bilateral). P stands for peel, R stands for *ROC*, and K-O stands for oil content in kernels.

**Figure 2 foods-12-00374-f002:**
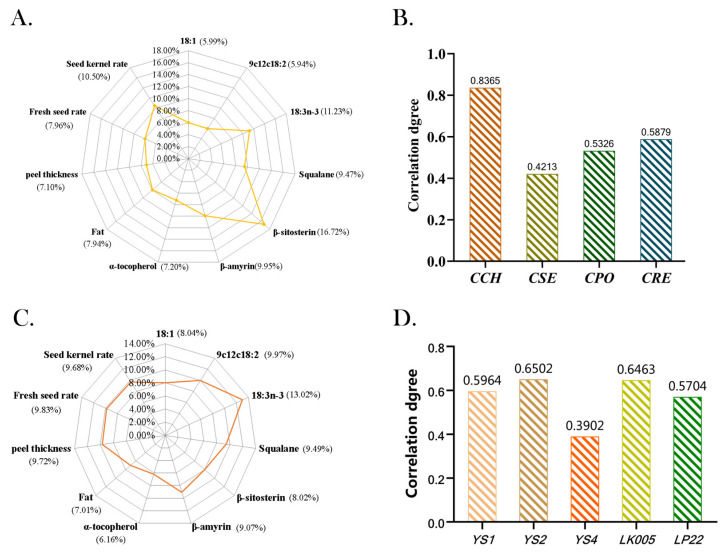
(**A**). The weights of nutrients in four *ROC* oils calculated according to the gray correlation coefficient method. (**B**). Correlation degrees of four *ROC* oils. (**C**). The weights of nutrients in five clones of *Camellia chekiangoleosa Hu.* oil, calculated according to the gray correlation coefficient method. (**D**). Correlation degree of five clones of *camellia chekiangoleosa Hu.* oil. YS-1, YS-2, YS-4, LK005 and LP22 all represent clone varieties of *Camellia chekiangoleosa Hu.,* and *CPO*, *CSE* and *CRE* fruit materials were mixed samples of 5 clones.

**Figure 3 foods-12-00374-f003:**
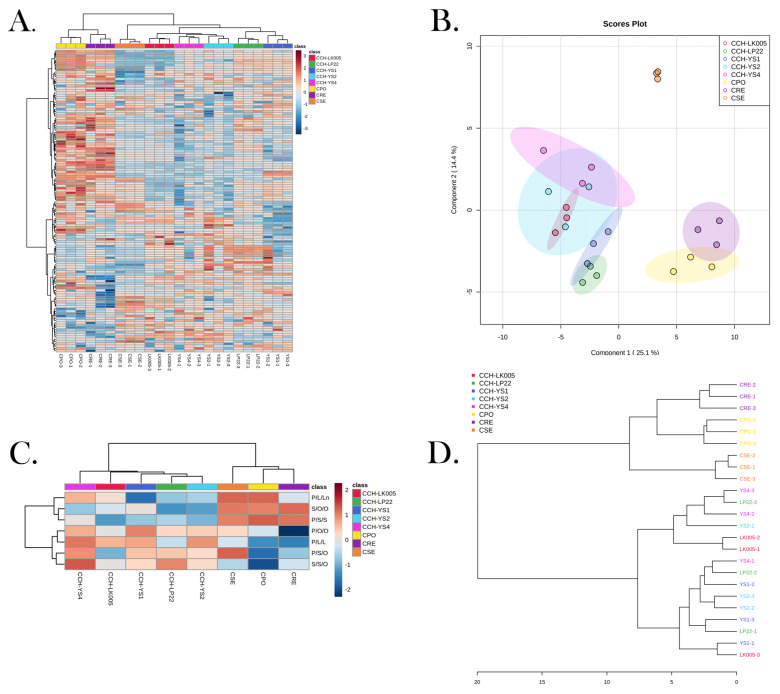
(**A**,**B**). Heatmap and PLSDA model diagram based on relative triglyceride content; (**C**,**D**). heatmap and dendrogram based on relative contents of characteristic triglycerides. YS-1, YS-2, YS-4, LK005 and LP22 all represent clone varieties of *ROC*, and *CPO*, *CSE* and *CRE* fruit materials were mixed samples of 5 clones.

**Table 1 foods-12-00374-t001:** Phenotype, fresh seed rate and kernel rate of four *ROC* fruits.

*ROC*		TD/mm	LD/mm	PT/mm	TD/LD	Fresh Seed Rate (%)	Seed Kernel Rate (%)
*C.chekiangoleosa Hu.*	YS-1	51 ± 0.24 ^a^	55 ± 0.90 ^b^	9 ± 0.42 ^b^	0.927	32.68 ± 0.10 ^d^	61.52 ± 0.31 ^b^
	YS-2	58 ± 0.36 ^e^	63 ± 0.54 ^d^	9 ± 0.72 ^b^	0.921	34.40 ± 0.16 ^e^	63.57 ± 0.41 ^d^
	YS-4	52 ± 0.92 ^b^	59 ± 0.62 ^c^	10 ± 0.50 ^c^	0.881	29.03 ± 0.17 ^c^	60.3 ± 0.56 ^a^
	Lk005	56 ± 0.72 ^d^	5 ± 0.40 ^c^	7 ± 0.46 ^a^	0.949	20.75 ± 0.49 ^a^	65.62 ± 0.68 ^e^
	LP22	52 ± 0.46 ^b,c^	53 ± 0.64 ^a^	9 ± 0.44 ^b^	0.981	22.86 ± 0.52 ^b^	62.19 ± 0.13 ^c^
*CCH*	mean	54 ± 0.54 ^B^	58 ± 0.62 ^B^	9 ± 0.51 ^A^	0.932	27.94 ± 0.29 ^D^	62.64 ± 0.42 ^D^
*C. semiserrata Chi.*	CSE	108 ± 0.42 ^D^	117 ± 0.40 ^D^	28 ± 0.14 ^D^	0.923	9.32 ± 0.34 ^A^	49.13 ± 0.25 ^A^
*C. polyodonta How ex. Hu.*	CPO	79 ± 0.52 ^C^	107 ± 0.82 ^C^	21 ± 0.98 ^C^	0.738	14.12 ± 0.26 ^B^	50.41 ± 0.45 ^B^
*C. reticulata Lindl.*	CRE	50 ± 0.10 ^A^	53 ± 0.74 ^A^	11 ± 0.30 ^B^	0.943	22.5 ± 0.33 ^C^	58.50 ± 0.74 ^C^

Note: YS-1, YS-2, YS-4, LK005 and LP22 all represent clone varieties of *Camellia chekiangoleosa Hu.*, and *CPO*, *CSE* and *CRE* fruit materials were mixed samples of 5 clones. LD and TD represent longitudinal diameter and transverse diameter, respectively. PT represents peel thickness. Different lowercase letters (a–e) represent significant differences between different clones of *Camellia chekiangoleosa Hu.*, and different uppercase letters (A–D) represent significant differences between different breeds of *ROC* (*p <* 0.05).

**Table 2 foods-12-00374-t002:** Chemical composition of camellia seed/peel depending on different species.

		*C. chekiangoleosa Hu.*					*CCH*	*C. semiserrata Chi.*	*C. polyodonta How ex. Hu.*	*C. reticulata Lindl*
Peel		YS-1	YS-2	YS-4	LK005	LP22	Mean	*CSE*	*CPO*	*CRE*
Fresh	Moisture (%)	71.60 ± 0.56 ^a^	76.07 ± 0.24 ^c^	79.86 ± 0.25 ^e^	77.45 ± 0.29 ^d^	73.60 ± 0.53 ^b^	75.72 ± 2.89 ^A^	80.59 ± 0.81 ^B^	82.10 ± 0.12 ^C^	74.87 ± 0.18 ^A^
Dried	Protein (mg/g)	47.24 ± 0.56 ^b^	27.41 ± 0.56 ^a,b^	25.53 ± 0.52 ^a^	17.58 ± 1.65 ^a^	30.61 ± 0.45 ^b^	29.67 ± 9.78 ^A,B^	33.6 ± 0.66 ^B^	22.6 ± 1.82 ^A^	34.3 ± 2.93 ^B^
	Soluble sugars (mg/g)	86.92 ± 1.71 ^c^	106.62 ± 2.82 ^d^	136.12 ± 3.39 ^f^	60.18 ± 5.04 ^a^	71.33 ± 1.99 ^b^	92.23 ± 26.93 ^C^	85.30 ± 1.45 ^B^	48.20 ± 4.06 ^A^	128.08 ± 3.24 ^D^
	Starch (mg/g)	125.71 ± 3.0 ^c,d^	104.85 ± 2.39 ^a^	128.95 ± 4.5 ^d,e^	117.54 ± 3.28 ^b^	132.41 ± 2.77 ^e^	121.89 ± 9.84 ^B^	119.62 ± 2.5 ^A^	127.59 ± 3.1 ^C^	118.61 ± 2.24 ^A^
	Oil (mg/g)	2.46 ± 0.051 ^a^	2.58 ± 0.066 ^a^	2.83 ± 0.044 ^a^	2.63 ± 0.079 ^a^	2.89 ± 0.032 ^a^	2.68 ± 0.16 ^A^	3.10 ± 0.042 ^A^	2.92 ± 0.073 ^A^	4.03 ± 0.044 ^B^
	FCR-reducing capacity	8.69 ± 0.49 ^a^	8.47 ± 0.08 ^a^	8.36 ± 0.49 ^a^	10.11 ± 0.22 ^b^	10.36 ± 0.24 ^b^	9.20 ± 0.86 ^C^	10.31 ± 0.15 ^C^	5.78 ± 0.21 ^B^	4.46 ± 0.09 ^A^
	Total flavonoids (mg/g)	4.32 ± 0.09 ^c^	2.61 ± 0.19 ^a^	2.39 ± 0.31 ^a^	3.92 ± 0.16 ^b,c^	5.47 ± 0.14 ^d^	3.74 ± 1.14 ^C^	5.32 ± 0.44 ^D^	3.40 ± 0.07 ^B^	1.94 ± 0.03 ^A^
Seed										
Fresh	Moisture (%)	21.97 ± 0.33 ^b^	20.84 ± 0.16 ^a^	20.66 ± 0.39 ^a^	23.32 ± 0.19 ^d^	22.57 ± 0.65 ^c^	21.87 ± 1.01 ^A^	37.71 ± 0.51 ^C^	45.33 ± 0.12 ^D^	34.80 ± 0.29 ^B^
Dried	Protein (mg/g)	95.56 ± 0.19 ^d^	92.51 ± 0.11 ^c,d^	90.06 ± 0.36 ^b,c,d^	75.35 ± 0.43 ^a^	105.86 ± 0.39 ^d^	91.87 ± 9.86 ^C^	76.81 ± 0.21 ^B,C^	71.03 ± 0.018 ^A^	75.92 ± 0.12 ^A,B^
	Soluble sugars (mg/g)	148.34 ± 9.50 ^b,c^	146.44 ± 4.55 ^b,c^	118.42 ± 3.07 ^a^	153.28 ± 4.03 ^c^	141.27 ± 2.59 ^b^	141.55 ± 12.19 ^B^	125.22 ± 2.32 ^A^	178.84 ± 6.15 ^d^	147.51 ± 5.88 ^b,c^
	Starch (mg/g)	97.33 ± 4.34 ^d^	74.56 ± 3.57 ^b,c^	61.86 ± 2.33 ^a^	73.12 ± 7.32 ^a,b^	67.77 ± 1.09 ^a,b^	74.93 ± 12.06 ^A^	75.37 ± 3.15 ^A^	84.37 ± 3.92 ^B^	105.41 ± 6.16 ^C^
	Oil (%)	56.12 ± 0.23 ^b,c^	56.44 ± 0.12 ^b,c^	57.19 ± 0.38 ^b,c^	53.61 ± 0.44 ^a,b^	58.84 ± 0.53 ^c^	56.44 ± 1.70 ^D^	50.49 ± 0.26 ^C^	39.13 ± 0.08 ^A^	43.69 ± 0.02 ^B^
	FCR-reducing capacity	8.59 ± 0.33 ^a,b^	9.21 ± 0.14 ^b,c^	8.82 ± 0.11 ^a,b,c^	9.13 ± 0.18 ^a,b,c^	9.35 ± 0.27 ^c^	9.02 ± 0.28 ^C^	7.37 ± 0.13 ^A^	7.02 ± 0.40 ^A^	8.46 ± 0.13 ^B^
	Total flavonoids (mg/g)	10.52 ± 0.49 ^a,b^	9.43 ± 0.38 ^a^	10.63 ± 0.14 ^a,b^	11.28 ± 0.42 ^b,c^	10.87 ± 0.73 ^a,b^	10.55 ± 0.62 ^A^	11.51 ± 0.85 ^B^	12.58 ± 0.57 ^C^	13.09 ± 0.14 ^D^

Note: YS-1, YS-2, YS-4, LK005 and LP22 all represent clone varieties of *Camellia chekiangoleosa Hu.*, and *CPO*, *CSE* and *CRE* fruit materials were mixed samples of 5 clones. Different lowercase letters (a–f) represent significant differences between different clones of *CCH*, and different uppercase letters (A–D) represent significant differences between different breeds of *ROC* (*p* < 0.05).

**Table 3 foods-12-00374-t003:** Lipid concomitants of the oil of four ROC oils.

	*C. chekiangoleosa Hu.*					*CCH*	*C. semiserrata Chi.*	*C. polyodonta How ex.Hu.*	*C. reticulata Lindl*
	YS-1	YS-2	YS-4	LK005	LP22	Mean	*CSE*	*CPO*	*CRE*
Squalane (mg/g)	0.72 ± 0.031 ^b^	0.67 ± 0.074 ^a^	0.66 ± 0.042 ^a^	0.70 ± 0.024 ^b^	0.71 ± 0.063 ^b^	0.69 ± 0.02 ^C^	0.43 ± 0.032 ^A^	0.47 ± 0.051 ^A^	0.59 ± 0.033 ^B^
β-Sitosterin (mg/g)	0.67 ± 0.064 ^b,c^	0.60 ± 0.10 ^a,b,c^	0.53 ± 0.073 ^a,b^	0.74 ± 0.057 ^c^	0.63 ± 0.082 ^a,b,c^	0.63 ± 0.07 ^B^	0.48 ± 0.012 ^A^	0.49 ± 0.066 ^A^	0.47 ± 0.048 ^A^
β-Amyrin (mg/g)	1.42 ± 0.41 ^b^	1.34 ± 1.43 ^b^	1.04 ± 1.88 ^a^	1.91 ± 3.30 ^c^	1.33 ± 3.05 ^b^	1.41 ± 0.28 ^C^	1.03 ± 0.80 ^B^	0.87 ± 0.67 ^A^	0.99 ± 0.97 ^B^
α-Tocopherol (mg/kg)	243.12 ± 1.11 ^b^	251.25 ± 1.24 ^c^	223.08 ± 0.38 ^a^	245.79 ± 0.39 ^b,c^	243.80 ± 0.48 ^b^	241.41 ± 9.60 ^B^	177.52 ± 0.97 ^A^	350.71 ± 0.63 ^C^	352.27 ± 0.31 ^C^
FCR-reducing capacity (ug/g)	162.79 ± 4.34 ^a,b,c^	158.25 ± 4.49 ^a,b^	156.46 ± 3.64 ^a^	168.34 ± 3.41 ^b,c^	171.65 ± 2.65 ^c^	163.50 ± 5.79 ^C^	130.72 ± 5.32 ^A^	164.82 ± 4.10 ^C^	157.41 ± 3.24 ^B^
Total flavonoids (ug/g)	428.61 ± 4.96 ^a^	437.38 ± 5.34 ^a,b^	435.26 ± 4.56 ^a^	440.37 ± 3.68 ^a,b^	449.27 ± 4.23 ^b^	438.18 ± 6.76 ^B,C^	353.49 ± 4.23 ^A^	437.65 ± 4.21 ^B,C^	429.04 ± 4.39 ^B^

Note: YS-1, YS-2, YS-4, LK005 and LP22 all represent clone varieties of *Camellia chekiangoleosa Hu.*, and *CPO*, *CSE* and *CRE* fruit materials were mixed samples of 5 clones. Different lowercase letters (a–c) represent significant differences between different clones of *CCH*, and different uppercase letters (A–C) represent significant differences between different breeds of *ROC* (*p* < 0.05).

**Table 4 foods-12-00374-t004:** Fatty acid composition of four *ROC* oils (area%).

	*C. chekiangoleosa Hu.*					*CCH*	*C. semiserrata Chi.*	*C. polyodonta How ex. Hu.*	*C. reticulata Lindl*
	YS-1	YS-2	YS-4	LK005	LP22	Mean	*CSE*	*CPO*	*CRE*
C14:0	0.049 ± 0.0061 ^a^	0.095 ± 0.019 ^b^	0.080 ± 0.0051 ^b^	0.087 ± 0.010 ^b^	0.081 ± 0.0076 ^b^	0.078 ± 0.016 ^B^	0.041 ± 0.0064 ^A^	0.084 ± 0.0076 ^B^	0.050 ± 0.0071 ^A^
C16:0	8.48 ± 0.092 ^a^	9.85 ± 0.013 ^c^	9.28 ± 0.069 ^b^	9.41 ± 0.12 ^b^	9.31 ± 0.040 ^b^	9.27 ± 0.44 ^C^	7.51 ± 0.036 ^A^	8.51 ± 0.0394 ^B^	10.88 ± 0.052 ^D^
C17:0	0.11 ± 0.013 ^a,b^	0.13 ± 0.002 ^b,c^	0.11 ± 0.0084 ^a,b^	0.16 ± 0.011 ^c^	0.11 ± 0.031 ^a,b^	0.12 ± 0.02 ^B^	0.058 ± 0.013 ^A^	0.11 ± 0.024 ^B^	0.057 ± 0.0047 ^A^
C18:0	3.96 ± 0.18 ^c,d^	3.22 ± 0.083 ^a^	4.37 ± 0.15 ^d^	4.05 ± 0.14 ^d^	3.49 ± 0.12 ^b,c^	3.82 ± 0.41 ^C^	3.25 ± 0.11 ^B^	2.47 ± 0.089 ^A^	3.40 ± 0.22 ^B^
SFA	12.60 ± 0.29 ^a^	13.30 ± 0.12 ^a,b^	13.84 ± 0.23 ^b^	13.71 ± 0.28 ^b^	12.99 ± 0.20 ^a^	13.29 ± 0.46 ^B^	10.86 ± 0.17 ^A^	11.17 ± 0.16 ^A^	14.39 ± 0.28 ^C^
10c C17:1	0.066 ± 0.014 ^a^	0.075 ± 0.017 ^a^	0.067 ± 0.0071 ^a^	0.10 ± 0.017 ^a^	0.088 ± 0.0037 ^a^	0.079 ± 0.013 ^A^	0.062 ± 0.0068 ^A^	0.083 ± 0.010 ^A^	0.064 ± 0.016 ^A^
9t C16:1	0.047 ± 0.018 ^a^	0.056 ± 0.014 ^a^	0.041 ± 0.010 ^a^	0.059 ± 0.0052 ^a^	0.042 ± 0.013 ^a^	0.049 ± 0.007 ^A^	0.031 ± 0.0025 ^A^	0.071 ± 0.019 ^A^	0.035 ± 0.0081 ^A^
9c C16:1	0.088 ± 0.020 ^cd^	0.086 ± 0.013 ^c^	0.073 ± 0.0082 ^a,b^	0.12 ± 0.0015 ^e,f^	0.052 ± 0.0046 ^a^	0.084 ± 0.022 ^C^	0.075 ± 0.0067 ^B^	0.069 ± 0.0077 ^A^	0.13 ± 0.0057 ^D^
9c C18:1	80.97 ± 0.13 ^c^	79.45 ± 0.066 ^a^	80.26 ± 0.18 ^b^	80.03 ± 0.23 ^b^	80.05 ± 0.095 ^b^	80.15 ± 0.49 ^C^	80.58 ± 0.048 ^C^	78.70 ± 0.14 ^B^	75.65 ± 0.26 ^A^
MUFA	81.17 ± 0.18 ^c^	79.67 ± 0.11 ^a^	80.44 ± 0.21 ^b^	80.31 ± 0.25 ^b^	80.23 ± 0.12 ^b^	80.36 ± 0.48 ^C^	80.75 ± 0.064 ^C^	78.92 ± 0.18 ^B^	75.88 ± 0.29 ^A^
9c C18:1/MUFA	99.75 ^a^	99.72 ^a^	99.78 ^a^	99.65 ^a^	99.78 ^a^	99.74 ± 0.048 ^A^	99.79 ^A^	99.72 ^A^	99.69 ^A^
C18:2 n-6	5.71 ± 0.039 ^c^	6.16 ± 0.058 ^d^	5.30 ± 0.047 ^a^	5.47 ± 0.048 ^b^	6.20 ± 0.017 ^d^	5.77 ± 0.36 ^A^	7.79 ± 0.11 ^B^	8.86 ± 0.079 ^C^	8.77 ± 0.14 ^C^
C18:3 n-3	0.52 ± 0.028 ^b^	0.88 ± 0.029 ^d^	0.42 ± 0.0075 ^a^	0.50 ± 0.024 ^b^	0.56 ± 0.036 ^b,c^	0.58 ± 0.16 ^A^	0.60 ± 0.028 ^B^	1.04 ± 0.029 ^C^	0.97 ± 0.032 ^C^
n-6/n-3	10.98 ^b^	7.00 ^a^	12.62 ^c^	10.94 ^b^	11.07 ^b^	10.52 ± 1.87 ^C^	12.98 ^D^	8.52 ^A^	9.04 ^B^
PUFA	6.23 ± 0.067 ^a,b^	7.04 ± 0.087 ^c^	5.72 ± 0.055 ^a^	5.97 ± 0.072 ^a^	6.76 ± 0.053 ^b^	6.34 ± 0.49 ^A^	8.39 ± 0.14 ^B^	9.90 ± 0.11 ^C^	9.74 ± 0.17 ^C^

Note: YS-1, YS-2, YS-4, LK005 and LP22 all represent clone varieties of *Camellia chekiangoleosa Hu.*, and *CPO*, *CSE* and *CRE* fruit materials were mixed samples of 5 clones. The unit of n-6/n-3 value is not a percentage. Different lowercase letters (a–f) represent significant differences between different clones of *CCH*, and different uppercase letters (A–D) represent significant differences between different breeds of *ROC* (*p* < 0.05).

## Data Availability

Data is contained within the article or Appendix A.

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
