# Peer review of "Comprehensive Evaluation of Quality Characteristics of Four Oil-Tea Camellia Species with Red Flowers and Large Fruit"

_foods, 2023, doi:10.3390/foods12020374_

Round 1

Reviewer 1 Report

My comments were sticky on attached pdf file

Reviewer 2 Report

This work is well written and investigated the fruit characteristics and the content of macronutrients and antioxidant substances in the peel and kernel of four Red-flowered oil-tea camellia (ROC).

I would like to suggest a few comments describe below.

Page 3, items 2.3 and 2.4.1:

It is necessary to include the references for both analyses.

Page 12 – Figure 3

it is not possible to visualize the results shown in this figure, redo it.

Page 13 - References

Line 495: There is no reference cited, please include it.

Line 522: Please complete this reference

Reviewer 3 Report

To the manuscript titled: “Comprehensive evaluation of quality characters of four oil-tea camellia with red flowers and large fruit” has scientific potential, however, several issues need to be explained/improved.

1. Materials

How the samples were dried and milled? Please provide details.

2. Determination of α-tocopherol

Something is wrong. The type of used column (Hypersil ODS2) does not fit to the used mobile phase (99.3% n-hexane and 0.7% isopropanol). Please provide an explanation/correction.

3. Total phenolic and …

Folin-Ciocalteu method has several meaningful disadvantages since the Folin–Ciocalteu reagent reacting not only with the phenolic compounds but also with some amino acids, peptides, reduction sugars, ascorbic acid, Maillard reaction products, and some more compounds, therefore several researchers’ advice to interpret obtained results as reducing capacity instead of total phenolics content. Please see for instance:

“The chemistry behind antioxidant capacity assays. J Agric Food Chem, 2005, 53, 1841–1856”.

 “Contribution of phenolic acids isolated from green and roasted boiled-type coffee brews to total coffee antioxidant capacity. European Food Research and Technology, 2016, 242, 641–653”.

4. Table 1. Phenotype, fresh seed rate and kernel rat four ROC.

“rat”?

Please provide an explanation between the seed and kernel of ROC to be clear for potential readers.

5. Conclusions

“This study showed that there are some links between fruit characteristics and oil quality, while the polyphenols in the peel also most likely play a role in protecting the oil from oxidation during the growth of ROC oil and ROC fruit with high TD/LD and thin peel thickness had higher oil yield and better oil quality, which was consistent with the result of grey correlation coefficient method.” – it is just speculation. Please remove.

“Moreover, through our determination, ROC oil had high edible value with higher content of oil and fat-soluble concomitant compared with Camellia sinensis oil and Camellia oleifera oil, reflected higher antioxidant value of ROC.” – “ROC oil had high edible value with higher content of oil” – bad sentence construction.

What do you mean by “higher dietary property”?

Round 2

Reviewer 3 Report

The manuscript was improved according to reviewers' comments and can be advised for publication after fixing small issues with the references. The references “[17] and [18]” are wrong described and cited. Please put attention to the author's name and surname's order and other detail e.g. please remove “[J]”. The same for reference “[20]”. In the reference “[12]” are over-used capital letters. Please put attention to such details.

Author Response

Thanks for your suggestion, please see the attachment.
